# HIV seropositive disclosure and associated factors among adult HIV positive clients in public health facilities in East Wollega Zone, Oromia Regional State, Western Ethiopia

Worku Fikadu[1]*, Chala Dechassa[2], Zelalem Desalegn[1], Adisu Ewunetu[1], Matiyos Lema[1], Adisu Tekle[1]

**1** Department of Public Health, Institute of Health Science, Wollega University, Nekemt, Ethiopia, **2** Gida Ayana Public Health Facility, Gida Ayana General Hospital, Ayana, Ethiopia

* wfikadu2@gmail.com, workufikadu8329@gmail.com

## Abstract

### Background

Disclosure of HIV-seropositive is important for HIV prevention and maintenance of health for people living with HIV and the community at large. Although HIV seropositive disclosure is mandatory for access to care and treatment, there is a paucity of data in the study setting.

### Objective

This study aimed to assess the magnitude of HIV Sero Positive Disclosure and associated factors among adult HIV positive clients in public health facilities in the east wollega zone, Oromia Regional State, Western Ethiopia, in 2023.

### Methods

A facility-based cross-sectional study design was conducted among 250 adult HIV-positive clients in public health facilities in East Wollega Zone. Systematic sampling was used to select the study participants. Data were collected from February 1 to *April 15, 2023*, using a structured, pretested, interviewer-administered questionnaire and reviewing patient cards using a checklist. Data were entered into Epi Data version 3.1 and exported to SPSS version 25 for analysis. Bivariable and multivariable logistic regression analyses were done. In bivariable analyses, variables with a P-value <0.25 were candidates for the multivariable model; in the multivariable model, significant associations were declared at a P-value < 0.05 and reported by an adjusted odds ratio along with a 95% confidence interval.

### Results

The magnitude of HIV seropositive disclosure to at least one person was found to be 68.2% (95%CI = 62.5%, 73.9%). HIV seropositive disclosure was significantly associated with being married ((AOR = 5.47, 95%CI: 2.87–10.43), low knowledge level (AOR = 0.095, 95%

**Data Availability Statement:** All relevant data are within the paper and its Supporting Information files.

**Funding:** The author(s) received no specific funding for this work.

**Competing interests:** The authors have declared that no competing interests exist.

**Abbreviations:** AIDS, Acquired Immunodeficiency Syndrome; AOR, Adjusted Odds Ratio; ARVs, Anti-Retrovirus; ART, Antiretroviral Therapy; CBHI, Community-Based Health Insurance; CDC, Communicable Disease Control; CD4, Cluster of Differentiation Four; CTC, Care and Treatment Center; COR, Crude Odd Ratio; EDHS, Ethiopian Demographic Health Survey; FMOH, Federal Ministry of Health; FHAPCO, Federal HIV/AIDS Prevention and Control Office; HCP, Health Care Provider; HIV, Human Immunodeficiency Virus; NACP, National AIDS Control Program; PLWHA, People Living With HIV/AIDS; PMTCT, Prevention of Mother-to-Child Transmission; UNAIDS, Joint United Nations Program on HIV AIDS; VCT, Voluntary Counseling and Testing; W H O, World Health Organization.

CI = 0.017–0.54), and perceived community discrimination (AOR = 0.192, 95% CI = 0.050, 0.74).

## Conclusion

This study reveals that HIV seropositive disclosure was low among the adult population living in the study area as compared to the other study findings. Disclosure of seropositives was significantly associated with marital status, knowledge level about the importance of disclosure, and perceived discrimination in societies. Healthcare providers and HIV prevention programme planners should provide health education about the importance of HIV disclosure and the stigma against HIV-positive patients by increasing society's awareness through appropriate channels.

## Introduction

### Background

HIV/AIDS is a global public health issue caused by two lentiviruses, HIV-1 and HIV-2 [1,2]. AIDS was first recognised in the 1980s and is the leading cause of death in developing countries, with over 40 million infected and two-thirds living in sub-Saharan Africa [3]. The World Health Organisation (WHO) and the United States Centres for Disease Control and Prevention (CDC) emphasise the importance of HIV status disclosure, particularly to sexual partners, as a crucial HIV prevention goal [4]. Disclosure is a difficult yet crucial decision for all infected individuals, but it also plays a significant role in coping with the disease [5]. In HIV prevention, "disclosure" is the process of revealing HIV-positive status to a sexual partner, family members, or others in their social circle [6]. While disclosure can have positive effects, such as increased social support and decreased stress, it can also have negative effects, depending on the recipient's response. For many HIV patients, disclosure can be stressful and emotionally charged and lead to personal anguish, loneliness, and social isolation [7–9]. Despite these challenges, HIV seropositive disclosure status plays a significant role in HIV prevention and management, facilitating greater social support and increased adherence.

In 2019, around 38 million people worldwide were living with HIV, with 67.3% in sub-Saharan Africa. In Ethiopia, 729,089 people were living with HIV [10–13]. HIV seropositive disclosure rates vary by location, subgroup of people living with HIV (PLWH), and potential confidante. In developed countries like the USA and Canada, disclosure is high, while in developing countries like Kenya and Sudan, it is between 60% and 70%. In Tanzania, 56.3% of HIV-positive patients disclosed their sero to their sexual partners, while in Uganda, only a small fraction of HIV-positive patients disclosed their sero to others [14]. In Ethiopia, 69% of HIV-positive women had disclosed to their sexual partners, while 73% of HIV-positive pregnant women had done so within four weeks of diagnosis.

Disclosure by PLWH is associated with less anxiety, fewer symptoms of depression, and increased social support. Failure to disclose HIV-positive status can expose close contacts to infection [6]. This study aimed to assess HIV seropositive disclosure among adult HIV-positive clients and identify context-based influencing factors for seropositive disclosure status [15–21].

Socio-demographic factors such as age, sex, occupation, marital status, and educational status were significantly associated with HIV seropositive disclosure [7,22–24]. Service-related

factors like ongoing counselling services and frequent follow-ups increased HIV seropositive disclosure [23–26]. Psychosocial factors such as fear of divorce, negative self-image, high perceived stigma, and discrimination decreased HIV seropositive disclosure [27–30]. HIV seropositive people run the danger of spreading the virus to their intimate partner and other family members who come into touch with them by hiding their positive status [18,19]. Even though transmission of HIV is increasing in the East Wollega Zone [21], there are limited studies conducted to assess the magnitude of HIV seropositive positive disclosure and its associated factors. The findings will help public health facilities and health professionals take the necessary interventions to increase HIV seropositive disclosure.

## Conceptual framework

This conceptual framework adapted from similar studies to show the influence of:-Sociodemographic factors, HIV-related factors, individual factors, psychosocial factors, and HIV seropositive disclosure [7,20,27,30] (**Fig 1**).

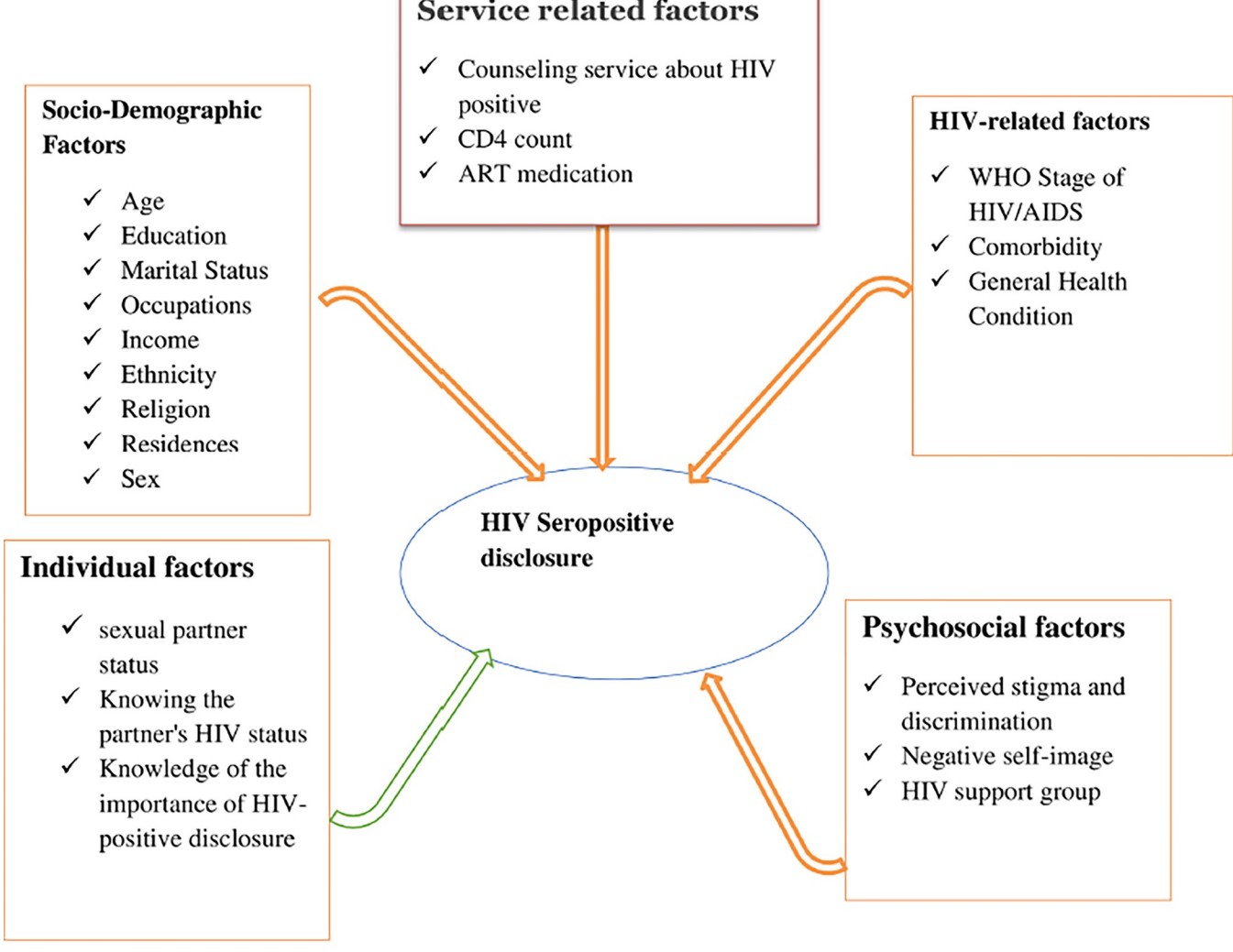

**Fig 1. Conceptual framework adapted after reviewing the literature.**

## Methods and materials

### Study area and study period

This study was conducted in East Wollega Zone Public Health Facilities. The Zone is located in western Ethiopia, 328 km away from the capital city Addis Ababa. It has 17 districts and, 43 towns and 287 rural kebeles. Based on the 2007 Census conducted by the CSA, this Zone has a total population of 1,213,503, of whom 606,379 are men and 607,124 women. The Zone has five Hospital, 327 health posts, 65 public health centers, one regional laboratory facility, as reported by the zonal Health Office. Only 10 public health facilities provide ART services and 50% of the whole ART site were randomly selected, these public health facilities include were Nekemt health center, Arjo hospital, Gida Ayana General Hospital, Guto Health Center, and Angar Gute Health Center with an average monthly attendant of 1400 ART follow-up patients. The study was conducted from February -1to April-15, 2023.

### Study design

A facility-based cross-sectional study design was employed to assess the magnitude of HIV Sero Positive Disclosure and associated factors among adult HIV Positive clients in public health facilities in East wollega Zone.

### Source population

All adult HIV-positive patients who were attending ART clinic at East wollega Zone public health facilities

### Study population

All randomly selected adult HIV-positive patients who were attending ART clinic at selected public health facilities in East Wollega Zone.

### Inclusion criteria

HIV positive Adults who were 18 years or above and attending ART services at selected public health facilities in East Wollega Zone were included from the study.

### Exclusion criteria

HIV-positive adult clients who are seriously ill and cannot respond to the survey during the study period.

### Sample size determination

The sample size was calculated using single population proportion formula. By using magnitude of HIV-sero positive disclosure from the study conducted at Mettu, in Ethiopia which was 69%, 95% confidence level and 5% degree of precision [9].

$$n = \frac{(Z\alpha/2)2p(1-p)}{d2} = 328.5 \sim 329$$

Since the Adult HIV-positive clients was lessthan10, 000 the following correction formula

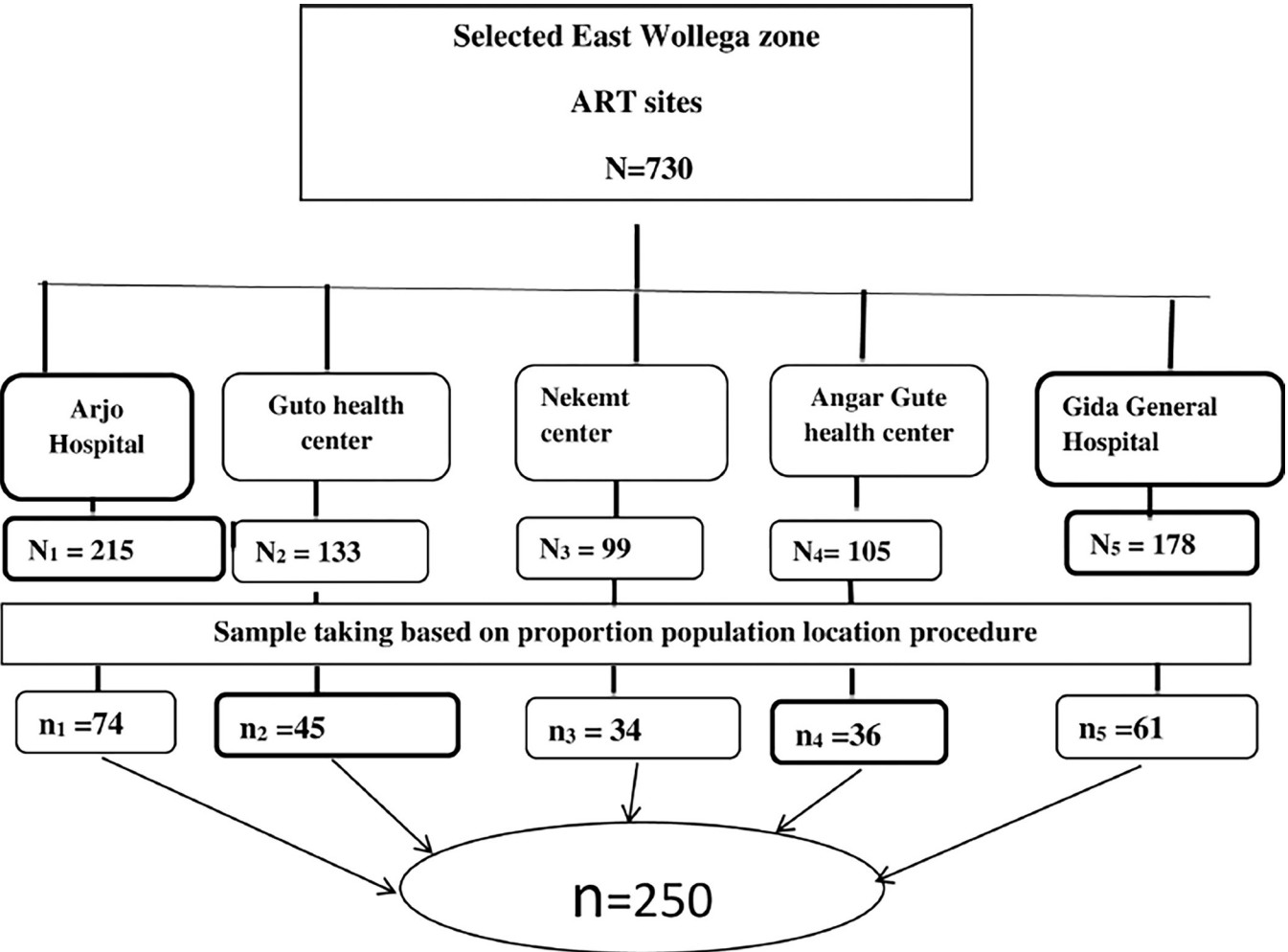

**Fig 2. Schematic presentation of sampling procedure for HIV sero positive disclosure and associated factors among adult HIV positive clients at East Wollega Zone public health facilities, Oromia, Ethiopia, 2023.**

was employed:

$$NF = n/(1 + (n/N)) = 329/(1 + (329/730))$$

$$= (\sim 227) \text{ by adding 10\% the final sample size was} = 249.68 \sim 250$$

## Sampling procedure

Gida Ayana General Hospital, Arjo Primary Hospital, and three health centres providing ART services in East Wollega Zone were included in this study. Currently, a total of 730 adults are attending these ART clinics. The final study units were selected by using systematic sampling techniques using a sampling interval (K). Since the total sample size for this study was 250, k = 730/250 = 2.92 = 3. The first client was selected randomly, and then every third client was included in the sample until the final sample size was achieved (Fig 2).

## Study variables

**Dependent variables.** HIV seropositive disclosure.

### Independent variables

- Socio-demographic factors: Age, Education, Marital status, Income, Occupation, Residence, Sex, and Ethnicity

- Illness factors: Stages of HIV AIDS, co-morbidity, General health condition,

- Individual factors:, discussion about HIV, knowledge on the importance of HIV positive disclosure,

- Psychosocial factors: Perceived stigma, discrimination factors, and negative self-image, HIV support group

- Services related factors:- ART medication, receiving counseling service, CD4 count,

### Operational definitions

**HIV seropositive disclosure**: The act of informing being an HIV positive to at least one person (sexual partner, parents, families, or friends [7].

HIV seropositive disclosure was measured by asking the question "Have you disclosed your HIV status to anyone?" and the response was dichotomized into Yes or No.

**Positive outcomes of disclosure**: are outcomes that facilitate or encourage an individual to disclose being HIV-positive [9].

The positive outcome of disclosure of HIV positive was measured by asking the question "If Positive outcome" and the response was categorized into four "Receiving kindness", "Acceptance", "Increased support", and "Decide to be tested for HIV".

**Negative outcomes of disclosure:**- Are those made to conceal HIV-positive status like stigma and rejection, divorce, and economic dependence [9].

The negative outcomes of disclosure were measured by asking the question " If negative outcome" and the response was categorized into four "Abandonment ", " Anger", " Blame", " Stigma"," Violence", and "Break up in the relationship".

### Data collection instrument and methods

Data were collected using a structured, interviewer-administered questionnaire and reviewing patient cards using a checklist. The data were collected by ten data collectors, including BSc. nurses and two supervisors (Msc. in public health), who were trained in HIV/AIDS comprehensive management and care. The training was given on data collection tools and data management.

### Data quality management

Data quality was maintained through regular communication and supervision of the data collection process. A questionnaire was pretested on 5% of the sample size at Sire General Hospital. The data were checked for completeness and cleaned before analysis.

### Data processing and analysis

Data was entered into Epi Data software version 3.1 and transported to SPSS version 25 for analysis. Descriptive statistics were used to summarise categorical variables, whereas means and standard deviation were used to summarise continuous variables. Hosmer and Lemeshow goodness of fit tests were used to check for model fitness, and the variance inflation factor was used to diagnose multi-collinearity between the explanatory variables with a cut-off point of

VIF > 10 as problematic. Binary logistic regression was carried out to assess factors associated with HIV seropositive Disclosure is expressed by the adjusted odds ratio (AOR) along with its respective 95% CI. Variables with a P value of less than 0.25 on the bi-variable level were entered into multivariable analysis. A P-value less than 0.05 was used to declare the level of significance.

## Ethics approval and consent to participate

The study was conducted in accordance with the Declaration of Helsinki and approved by Wallaga University's Institutional Review Board (Protocol number WU/RO/553/2014) and permission to conduct the study in health facilities was secured from zonal health department and respective health facilities. Detailed explanation to the study participants on the fact that data collection procedure has no any harm to them and other community and it was used only for research purpose. The participants provided written consent, and Confidentiality was maintained. The findings were presented as aggregate and would not reveal respondents' identities.

## Results

### Socio-demographic characteristics of the participants

A total of 238 ART patients participated in this study, giving a response rate of 95.2%. The mean (±SD) age of the study participants was 30.12 (±8.37) years. One hundred twenty (50.4%) were female. More than half of the one hundred forty-three (60.1%) respondents were urban residents. Regarding marital status, one hundred thirty-eight (58.0%) were married. Regarding the educational status, ninety-seven (40.8%) of the study participants completed secondary school. Regarding income, ninety (37.8%) were less than 1500 monthly income (Table 1).

**HIV-related factors.** From the total respondents, 145 (60.9%) had WHO stage I at ART start. Of the total respondents, 134 (56.5%) had comorbidity conditions (Table 2).

### Individual factors

From a total of 238 respondents, 160(67.2%) had sexual partners, among those who had sexual partners, 95 (59.4%) disclosed their HIV positive status to their partners, among these, 110 (67.4%) had positive outcomes reactions from anyone else. The majority of the respondents 205(86.0%) were knowledgeable about the importance of someone's HIV-positive disclosure (Table 2).

### Service related factors

From the respondents, 140 (85.8%) received counselling for HIV positive disclosure. From a total of 238 respondents, 125 (53.6%) had a CD4 count greater than 250 at ART start. From a total of 169 (71.0%) recent CD4 count greater than 250, 84 (35.3%) of respondents were reported to have been on ART for the duration of 1–2 years (Table 2).

**Psychosocial factors.** Among the study participants, 190 (79.8%) were not community discriminated against PLWHA, 10 (20.3%) had a low level of perceived discrimination, and 144 (60.5%) had no negative self-image. Of a total of respondents, 130 (79.7%) had an HIV support group (Table 2).

**Table 1. Socio-demographic characteristics of respondents among HIV positive adults in East Wollega Zone public health facilities, Oromia, Ethiopia, 2023 (n = 238).**

| Variables | Category | Frequency | Percent |
|---|---|---|---|
| Age | 18–24 | 67 | 28.2 |
| | 25–34 | 102 | 42.9 |
| | 35–44 | 54 | 22.7 |
| | > = 45 | 15 | 6.3 |
| Sex | Male | 118 | 49.6 |
| | Female | 120 | 50.4 |
| Residence | Urban | 143 | 60.1 |
| | Rural | 95 | 39.9 |
| Marital status | Single | 81 | 34.0 |
| | Married | 138 | 58.0 |
| | Divorced | 7 | 2.9 |
| | Widowed | 12 | 5.0 |
| Educational status | No formal education | 34 | 14.2 |
| | Primary school | 28 | 11.7 |
| | Secondary school | 97 | 40.7 |
| | College and above | 79 | 33.1 |
| Occupation | Housewife | 24 | 10.1 |
| | Farmer | 24 | 10.1 |
| | Daily laborer | 51 | 21.4 |
| | Government Employee | 60 | 25.2 |
| | NGO | 11 | 4.62 |
| | Merchant | 29 | 12.2 |
| | Student | 39 | 16.4 |
| Ethnicity | Oromo | 148 | 62.2 |
| | Amara | 65 | 27.7 |
| | Gurage | 15 | 6.3 |
| | Tigre | 9 | 3.8 |
| Religion | Protestant | 64 | 26.9 |
| | Orthodox | 114 | 47.9 |
| | Muslim | 52 | 21.8 |
| | Catholic | 8 | 3.4 |
| Monthly income | <1500 | 90 | 37.8 |
| | 1500–3000 | 72 | 30.3 |
| | 3001–5000 | 50 | 21.0 |
| | >5000 | 26 | 10.9 |

## Magnitude of HIV Sero Positive Disclosure

Magnitude of HIV Sero Positive Disclosure was found to be 68.2% (95%CI = 62.5%-73.9%) (Fig 3).

## Multivariable logistic regression

Multivariable analysis conducted done to identify factors significantly associated with HIV seropositive disclosure. Those variables showed an association with HIV seropositive disclosure p-value < 0.25 on the bivariable model were selected as candidate variables for multivariable analysis. The result of multivariable analysis shows, married participants were 5.4 times

**Table 2. Distribution of factors related to HIV status disclosure among HIV positive adults in East Wollega Zone facilities, Oromia region, Ethiopia,2023(n = 238).**

| Variables | Category | n | % |
|---|---|---|---|
| **HIV related factors** | | | |
| WHO stage at ART start | I | 145 | 60.9 |
| | II | 29 | 12.2 |
| | III | 60 | 25.2 |
| | IV | 4 | 1.6 |
| Comorbidity conditions | Yes | 134 | 56.5 |
| | No | 104 | 43.5 |
| **General health condition** | **Worked** | **224** | **94.1** |
| | **Ambulatory** | **14** | **5.90** |
| **Individual factors** | | | |
| **Have sexual partner** | **Yes** | **160** | **67.2** |
| | **No** | **78** | **32.8** |
| **Disclose HIV seropositive to a partner** | **Yes** | **95** | **59.4** |
| | **No** | **65** | **40.6** |
| **Not disclose your HIV seropositive to your partner** | **Fear of abandonment** | **26** | **40.0** |
| | **Fear of confidentiality** | **30** | **46.1** |
| | **Fear of accusation of infidelity** | **9** | **14.0** |
| Reaction from anyone else | **Positive outcome** | **110** | **67.4** |
| | **Negative outcome** | **53** | **32.5** |
| **Positive outcome** | **Receiving kindness** | **30** | **27.2** |
| | **Acceptance** | **20** | **18.4** |
| | **Increased support** | **60** | **54.5** |
| **Negative outcome** | **Anger** | **7** | **13.2** |
| | **Blame** | **8** | **15.1** |
| | **Stigma** | **18** | **33.9** |
| | **Break up in the relationship** | **20** | **37.7** |
| **Knowledge on the importance of someone's HIV-positive disclosure** | **Yes** | **205** | **6.0** |
| | **No** | **33** | **14.0** |
| Importance of disclosing someone's HIV-positive | **Receive emotional release** | **53** | **25.8** |
| | **Felt a perceived duty to inform partner** | **37** | **18** |
| | **Access medical and financial support** | **108** | **52.7** |
| | **Reduce the stress of the infected Partner** | **7** | **3.41** |
| **Service related factors** | | | |
| **Duration of ART medication** | Less than 1 year | 65 | 27.3 |
| | 1–2 year | 84 | 35.3 |
| | 3–4 years | 28 | 11.8 |
| | More than 4 years | 61 | 25.6 |
| CD4 Count at ART start (cells/ml) | <250 | 113 | 47.4 |
| | ≥250 | 125 | 53.6 |
| **Psychosocial factors** | | | |
| Community-discrimination against PLWHA | Yes | 48 | 20.2 |
| | No | 190 | 79.8 |
| **Not-discriminate-against PLWHA** | **Received education on HIV** | **64** | **26.9** |
| | **Most people have been affected by HIV** | **60** | **25.2** |
| | **I have never experienced any** | **114** | **47.9** |
| **level of perceived discrimination** | **Low** | **10** | **20.3** |
| | **High** | **38** | **79.7** |

*(Continued)*

**Table 2.** (Continued)

| Variables | Category | n | % |
|-----------|----------|---|---|
| **Negative self-image** | **Yes** | **94** | **39.5** |
| | **No** | **144** | **60.5** |
| **HIV support group** | **Yes** | **130** | **79.7** |
| | **No** | **33** | **20.3** |

(AOR = 5.47, 95%CI: 2.87–10.43) more likely to disclose their HIV seropositive compared to Unmarried ones. People who were unaware of the importance of HIV Positive disclosure were 90.5% (AOR = 0.095, 95%CI: 0.017–0.54) less likely to disclose their HIV status. Those who perceived community discrimination of their HIV sero positive disclosure status were 80.8% (AOR = 0.192, 95%CI: 0.05–0.74) less likely to disclose their HIV sero positive disclosure status (Table 3).

## Discussion

This study showed that out of 238 participants, 68.2% (95% CI = 62.5%–73.9%) had disclosed their HIV seropositive status at least to one person. This finding was lower than similar studies from Tanzania (79.8%) [23], Kenya (80%), and Nigeria (97.5%) [25,31]. The finding was also lower than the findings from studies conducted in Axum Health Facilities (80.1%) [18], Ambo Hospital (86.2%) [20], and Debre Markos Town (96.4%) [8] in Ethiopia. A further finding from a systematic review conducted in Ethiopia to assess disclosure of HIV seropositivity to sexual partners showed it was also higher compared to the current result from the study (76.3%) [7]. This could be due to the disparity between ART service delivery systems and their respective economies, cultures, and social structures. However, this finding was higher than the findings of similar studies in Sarawak, Malaysia (63%) [22]. Similarly, this result was higher in comparison to the findings from Ethiopia at Mekelle Hospital (63.8%) [19], and public health institutions in Dire Dawa (60.6%) [2].

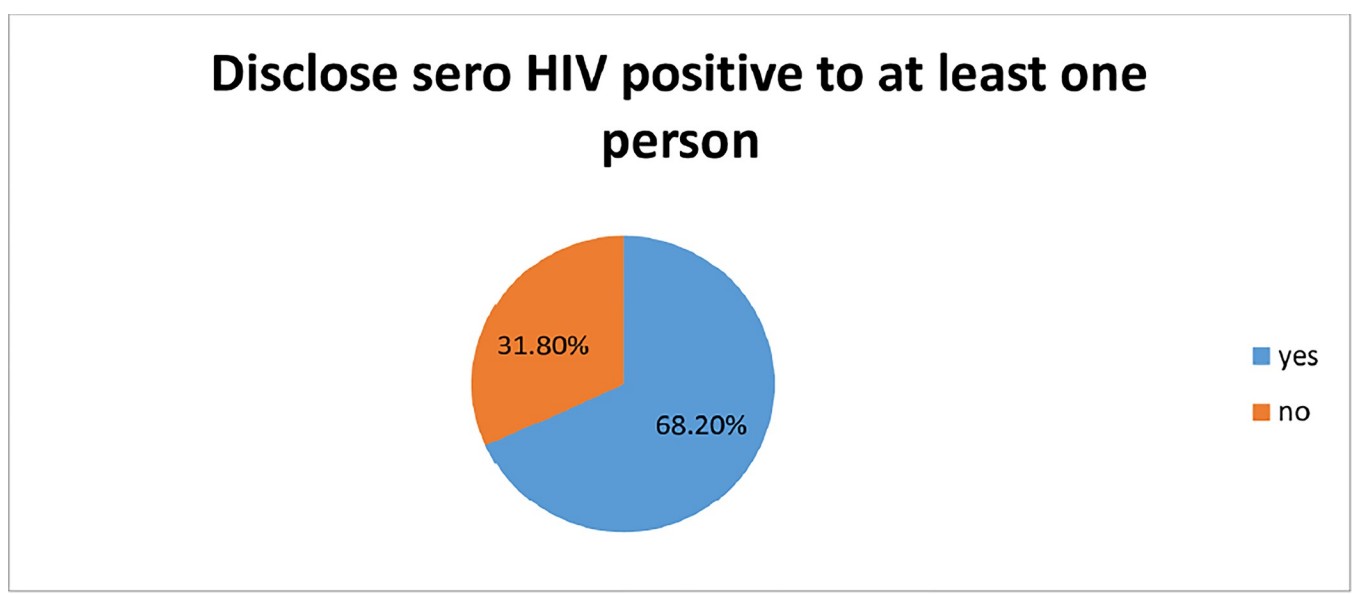

**Fig 3. Magnitude of HIV Sero Positive Disclosure status among adults attending ART clinics in East Wollega Zone.**

**Table 3. Multi-variable logistic regression analysis of factors associated with HIV positive disclosure among Adult HIV Positive clients in public health facilities in East wollega zone.**

| Variables | Category | HIV Disclosure | | AOR(95% CI) | P-value |
|---|---|---|---|---|---|
| | | Yes | No | | |
| Sex | Male (ref) | 50 | 68 | 1 | |
| | Female | 90 | 3027 | 4.08(1.0–6.22) | 0.0524 |
| Age | 18–24 | 40 | | 1 | |
| | 25–34 | 81 | 21 | 2.6(1.3–5.16) | 0.73 |
| | 35–44 | 34 | 20 | 1.1(0.54–2.39) | 0.30 |
| | > = 45 | 8 | 7 | 0.7(0.25–2.37) | 0.06 |
| Residence | Urban | 92 | 51 | 1 | |
| | Rural | 71 | 24 | 1.63(0.92–2.91) | 0.178 |
| Marital status | Single | 42 | 39 | 1 | |
| | Married | 118 | 20 | 5.47(2.87–10.43) | 0.001* |
| | Divorced/widowed | 3 | 16 | 0.15(0.01–1.34) | 0.38 |
| Educational status | No formal education | 18 | 16 | 0.25(0.10–1.03) | 0.56 |
| | Primary school | 20 | 8 | 1.09(0.1–10.46) | 0.60 |
| | Secondary school | 63 | 34 | 0.6(0.25–1.82) | 0.92 |
| | College and above | 62 | 17 | 1 | |
| Monthly income | <1500 | 50 | 40 | 0.46(0.17–1.20) | 0.11 |
| | 1500–3000 | 51 | 21 | 0.89(0.32–2.44) | 0.17 |
| | 3001–5000 | 43 | 7 | 2.26(0.69–7.35) | 0.67 |
| | >5000 | 19 | 7 | 1 | |
| Comorbidity conditions | Yes | 107 | 27 | 3.32(0.87–5.90) | 0.051* |
| | No | 56 | 47 | 1 | |
| Received counseling to HIV positive disclosure status | Yes | 146 | 57 | 1 | |
| | No | 18 | 17 | 0.43(0.20–0.58) | 0.015 |
| HIV support group | Yes | 140 | 52 | 1 | |
| | No | 23 | 23 | 0.37(0.19–1.71) | 0.75 |
| Knowledge on the importance HIV-positive disclosure | Yes | 152 | 53 | 1 | |
| | No | 11 | 21 | 0.095(0.017–0.54) | 0.008* |
| Community discrimination against PLWHA | Yes | 26 | 22 | 0.192(0.05–0.74) | 0.017* |
| | No | 137 | 53 | 1 | |
| General Health Status | Worked | 155 | 69 | 1 | |
| | Ambulatory | 8 | 6 | 0.59(0.19–1.77) | 0.05 |

This study found a strong relationship between HIV disclosure status and marital status, understanding of the significance of seropositive disclosure, and perception of the discrimination of seropositive people in the community. In this study, married participants were 5.4 times more likely to disclose their HIV seropositive status compared to unmarried participants (AOR = 5.47, 2.87–10.43) (Table 3). This finding was inconsistent with a similar study conducted in the Kilombero district of southeastern Tanzania among adults attending care and treatment clinics [12,23].

This finding contradicts the result of the study conducted among an Iranian woman and a patient living with HIV which showed women who were married or had cohabiting sexual partners were less likely to disclose their HIV seropositive status [6,9]. This might be due to the lack of a culture of common decision-making in marriage among the Iranian community compared to the Ethiopian community [6,9]. This study found that people who were unaware of the importance of HIV disclosure were 90% less likely to disclose their HIV seropositive

status (AOR = 0.095, 95%CI: 0.017–0.54). The result is in line with the findings of a study conducted among adult HIV-positive clients at a tertiary hospital in the Niger Delta in Tanzania [15,27]. The study showed that those who perceived community discrimination based on their HIV status were 80% less likely to disclose their HIV status (AOR = 0.192, 95% (AOR = 0.192, 95%CI: 0.05–0.74) (Table 3). In a similar study conducted in African countries among patients receiving HIV care in Kabale, Uganda, Michael [6,16,17], and people living with HIV/AIDS in Ogun State, Nigeria [5], and Kilombero district, South-Eastern Tanzania [18], ART patients with perceived stigma were less likely to disclose their HIV-positive status than those who didn't perceive stigma. This is further supported by findings from studies at public health Facilities in Butajira Town and Bale Zonal Hospitals in Ethiopia [20,30].

A possible justification is that HIV-related stigma leads to non-disclosure of HIV status and that stigmatised patients fear the consequences of disclosure, including depression, social withdrawal, psychological distress, and loss of family support [6,9]. Fear of discrimination, fear of breach of confidentiality, and fear of family members make them hide their status [24].

## Strength of the study

The response rate of the study participant in the area was good enough and Card revision was conducted to check for against self-reported response.

## Limitations of the study

This study can be improved by adding qualitative methods to explore the reasons behind non-disclosure. Since the cross-sectional study, the study doesn't show cause-and-effect relationships. This study was based on self-reporting of the disclosure status which might overestimate the outcome variable because of social desirability.

## Conclusion

According to this study, few adult HIV-positive patients in East Wollega Zone disclose their status. The disclosure status was substantially associated with marital status, knowledge of the significance of HIV seropositive disclosure, and perceptions of community discrimination towards seropositive people.

## Recommendations

Based on the finding of the study the following recommendations are important to increase the importance of HIV-positive disclosure among adults in East Wollega Zone.

### For health professionals

- Should strengthen the provision of information and education about HIV-positive disclosure status.

### For the zonal health department

- Should work to raise the knowledge of the importance of disclosure for HIV-positive adults by designing proper health education through different programmes such as HIV-positive conferences regularly, via local media to enhance the importance of disclosure among HIV-positive adults.

- The zonal health department should advocate for HIV patients known against community-based discrimination.

- The zonal health department has to invite governmental and nongovernmental sectors to give health education and training to HIV-positive adults.

- The zonal health department should link HIV clients with higher perceived discrimination to respective Support group.

### For future researchers

- Longitudinal study and qualitative study needs to be done to identify important factor and better understanding of importance of the HIV positive disclosure.

## Supporting information

**S1 File. Collected to assess magnitude of HIV sero positive disclosure and associated factors among adult HIV positive clients at East Wollega Zone public health facilities, Oromia, Ethiopia, 2023.**
(XLSX)

**S1 Dataset. Data cleand for assessement of HIV sero positive disclosure and associated factors among adult HIV positive clients at East Wollega Zone public health facilities, Oromia, Ethiopia, 2023.**
(SAV)

## Acknowledgments

First of all, we would like to thank Wallaga University for providing us this opportunity of conducting this research. We would also like to thank study participants, data collectors, officials of East Wollega zonal health department and focal persons of all Public Health Facilities, where the study was conducted in.

## Author Contributions

**Conceptualization:** Worku Fikadu, Chala Dechassa.

**Data curation:** Worku Fikadu.

**Formal analysis:** Worku Fikadu, Matiyos Lema, Adisu Tekle.

**Investigation:** Worku Fikadu.

**Methodology:** Worku Fikadu, Chala Dechassa, Zelalem Desalegn, Adisu Ewunetu, Matiyos Lema, Adisu Tekle.

**Supervision:** Adisu Ewunetu, Adisu Tekle.

**Validation:** Chala Dechassa.

**Writing – original draft:** Worku Fikadu, Chala Dechassa, Zelalem Desalegn, Adisu Ewunetu, Matiyos Lema, Adisu Tekle.

**Writing – review & editing:** Worku Fikadu, Chala Dechassa, Zelalem Desalegn, Adisu Ewunetu, Matiyos Lema.

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
