## [Decision Letter · Decision Letter 0]

12 Apr 2024

PONE-D-23-42840HIV SERO POSITIVE DISCLOSURE AND ASSOCIATED FACTORS AMONG ADULT HIV POSITIVE CLIENTS IN PUBLIC HEALTH FACILITIES IN EAST WOLLEGA ZONE, OROMIA REGIONAL STATE, WESTERN ETHIOPIA.PLOS ONE

Dear Dr. FIKADU,

Thank you for submitting your manuscript to PLOS ONE. After careful consideration, we feel that it has merit but does not fully meet PLOS ONE’s publication criteria as it currently stands. Therefore, we invite you to submit a revised version of the manuscript that addresses the points raised during the review process.

The article contains a lot of grammatical error which should be corrected. Ensure to consider comments from both reviewers. 

We look forward to receiving your revised manuscript.

Kind regards,

Moses Katbi MD, MPH, MBA, DrPH

Academic Editor

PLOS ONE

10.11694/pamj.supp.2014.18.1.3551

file:///home/nkw-ld22-090/Downloads/PONE-D-23-42840.pdf

In your revision ensure you cite all your sources (including your own works), and quote or rephrase any duplicated text outside the methods section. Further consideration is dependent on these concerns being addressed.

5. Please remove your figures from within your manuscript file, leaving only the individual TIFF/EPS image files, uploaded separately. These will be automatically included in the reviewers’ PDF.

Reviewers' comments:

Reviewer's Responses to Questions

**Comments to the Author**

1. Is the manuscript technically sound, and do the data support the conclusions?

Reviewer #1: Partly

Reviewer #2: No

2. Has the statistical analysis been performed appropriately and rigorously? 

Reviewer #1: Yes

Reviewer #2: No

3. Have the authors made all data underlying the findings in their manuscript fully available?

Reviewer #1: Yes

Reviewer #2: No

4. Is the manuscript presented in an intelligible fashion and written in standard English?

Reviewer #1: No

Reviewer #2: Yes

5. Review Comments to the Author

Reviewer #1: 1. How can the authors assure the nobility of this study?

2. Since the study involved more than one health facility, the sampling method should pass at least two stages and authors have to use design effect for computing sample size

3. the structure of manuscript should follow the authors instruction of this journal

4. the discussion section should be complied in detail rather than comparison of findings

5. grammatical errors have to checked and reviewed by native speakers or approved grammar checkers

Reviewer #2: The study is very interesting with historical significance. It is very informative and relevant to the advancement of blended learning method.

However, the use of data and statistics to support the conclusions is sparse.

This study uses qualitative research method, but there was so many missing information about the data collection method, process and analysis. The study stated that eighty-two students were recruited, but the selection criteria were not clearly stated. Also, the demography of the participants was not included. These are important piece of information that can help the understanding of the dynamics of the participants and the effect on the finding and conclusions. The data analysis was inadequately presented, and it is difficult to see how the conclusions were data driven. The use of tables and charts to present the results will improve the quality of the result.

It will also be very helpful to organize the study under the main sections (introduction, methodology, result and discussion.

The ethical statement shows that the consents were taken verbally, and these verbal consents were documented. how were they documented? It will be a significant additional information to the ethical section.

Finally, the team should review the grammar and the structure of the writing to improve the readability of the article.

6. PLOS authors have the option to publish the peer review history of their article (what does this mean?). If published, this will include your full peer review and any attached files.

Reviewer #1: No

Reviewer #2: No

---

## [Author Response · Author response to Decision Letter 0]

24 May 2024

Author’s response to the reviewers and editor

Title:

HIV seropositive disclosure and associated factors among adult HIV positive clients in public health facilities in East Wollega Zone,Oromia Regional State, Western Ethiopia.

Authors:

Worku Fikadu (wfikadu2@gmail.com)

Chala Dechasa (chala124dechasa@gmail.com)

Zelalem Desalegn (zolad09@gmail.com)

Adisu Desisa (adisuewunetu2019@gmail.com)

Matiyos Lema (maatii3399@gmail.com)

Adisu Tekle (adisnuro7@gmail.com)

Date: April 18, 2024

Author’s response to the reviewers:

Firstly, authors express their appreciation to the reviewer. We believe that their vital comments and suggestions have substantially improved the presentation of our study, as well as its overall quality and the manuscript. Following, we offer a point-by-point response to the issues and points the reviewers raised regarding the original manuscript.

Reviewer #1

Comment#1. How can the authors assure the nobility of this study?

Authors response:This study is noble due to the difference in sociodemographic and behavioural characteristics of the current study population. Even though HIV positive disclosure status has increased in Ethiopia, the disclosure status of the east wollega zone is unknown, and the magnitude of the HIV transmission is high. The authors hypothesise that the HIV transmission rate might be due to low disclosure of HIV positive status, as HIV seropositive people run the danger of spreading the virus to their intimate partner and other family members who come into contact with them by hiding their positive status. Hence, the authors were interested in knowing the magnitude of HIV seropositive status disclosure and the factors associated with disclosure.

2. Since the study involved more than one health facility, the sampling method should pass at least two stages and authors have to use design effect for computing sample size

Author’s response: we appreciate the comment.

The authors identified 10 ART-provider clinics in East Wollega and then selected 50% of the clinics randomly. Finally, the authors allocated study units proportionally to the selected health facilities. We initially considered the design effect, but after discussion, the authors believed that increasing the sample size did not improve the precision of the population estimate. We know that the design effect quantifies the extent to which the expected sampling error in a survey departs from the sampling error that can be expected under simple random sampling. Since we took the adult HIV patient population attending in 50% of clinics, we believed the sampling error in our selected population would not be large. However, we are happy that you suggested the right thing.

3. the structure of manuscript should follow the authors instruction of this journal

Authors-Response: We revised the manuscript according to the journal instructions following (IMRAD); the figures and tables were removed from the original manuscript and submitted in TIFF format separately. We also followed the font size recommendation of the journal.

4. the discussion section should be complied in detail rather than comparison of findings

Authors response: The discussion part was revised was revised by authors in four different paragraph

The authors provided essential interpretation to the key findings.

secondly the authors compare and contrast to previous studies

We discussed unexpected(contradictory findings)

Thirdly we highlighted the significance of the findings 

lastly we discussed unanswered questions and potential future research

5. grammatical errors have to checked and reviewed by native speakers or approved grammar checkers

Authors response: We have gone through the manuscript and found typographic and grammatical errors, and we have revised the manuscript extensively to improve its readability.

Comments by reviwer#2

Reviewer #2: The study is very interesting with historical significance. It is very informative and relevant to the advancement of blended learning method. However, the use of data and statistics to support the conclusions is sparse. This study uses qualitative research method, but there was so many missing information about the data collection method, process and analysis. The study stated that eighty-two students were recruited, but the selection criteria were not clearly stated. Also, the demography of the participants was not included. These are important piece of information that can help the understanding of the dynamics of the participants and the effect on the finding and conclusions. The data analysis was inadequately presented, and it is difficult to see how the conclusions were data driven. The use of tables and charts to present the results will improve the quality of the result.It will also be very helpful to organize the study under the main sections (introduction, methodology, result and discussion.

Author response: we appreciate the comment; We prepared the manuscript using IMRAD format 

However we doubt that the comments seems for other manuscript than ours.

Possible justifications

1:A. The Authors didn’t use qualitative research method

2; The study stated that eighty-two students were recruited, but the selection criteria were not clearly stated, However we didn’t recruit students our sample was from HIV positive patients.

Hence dear reviewer, please make sure that you have reviewed that our manuscript, then we will use to improve the presentation 

The ethical statement shows that the consents were taken verbally, and these verbal consents were documented. how were they documented? It will be a significant additional information to the ethical section. 

Author’s response: we appreciate the comment and it has been included as follows.

Ethics approval and consent to participate: The study was conducted by the Declaration of Helsinki and approved by Wallaga University's Institutional Review Board (Protocol number WU/RO/553/2014) and permission to conduct the study in health facilities was secured from zonal health department and respective health facilities. Detailed explanation to the study participants on the fact that data collection procedure has no any harm to them and other communities and it was used only for research purposes. The participants provided written consent, and Confidentiality was maintained. The findings were presented as aggregate and would not reveal respondents' identities.

Finally, the team should review the grammar and the structure of the writing to improve the readability of the article.

Author’s response: We have gone through the manuscript and found typographic and grammatical errors, and we have revised the manuscript extensively to improve its readability.

---

## [Editor Report · Decision Letter 1]

29 May 2024

HIV seropositive disclosure and associated factors among adult HIV positive clients in public health facilities in East Wollega Zone,Oromia Regional State, Western Ethiopia.

PONE-D-23-42840R1

Dear Fikadu Worku,

We’re pleased to inform you that your manuscript has been judged scientifically suitable for publication and will be formally accepted for publication once it meets all outstanding technical requirements.

Kind regards,

Moses Katbi

Academic Editor

PLOS ONE